# Assessment of patient safety culture and associated factors among healthcare professionals in public hospitals of Bahir Dar City, Northwest Ethiopia: A mixed-methods study

**Daniel Atinafu ⬤*, Gebremariam Getaneh, Getachew Setotaw**

Department of Health Systems Management and Health Economics, College of Medicine and Health Science, School of Public Health, Bahir Dar University, Bahir Dar, Ethiopia

* atinafudaniel23@gmail.com

## Abstract

### Background

Patient safety is an essential component of healthcare quality. Despite enormous advances in medical knowledge, many adverse events continue to endanger patient safety. Although mixed-method studies are necessary to gain a deeper understanding of safety culture, few studies provide practical evidence of patient safety culture and associated factors in Ethiopia. This study aimed to assess patient safety culture and associated factors among healthcare professionals in public hospitals in Bahir Dar City, Northwest Ethiopia.

### Methods

A cross-sectional study design was employed, in triangulation with qualitative methodologies, from March 10 to April 10, 2022. A stratified sampling technique was used to select 420 study participants from three public hospitals. A standardized tool measuring 12 patient safety culture composites was used for data collection. Purposive sampling was employed in the qualitative study. Bi-variable and multivariable linear regression analyses were performed using SPSS version 23, with significance set at a 95% confidence interval and a p-value of <0.05. Content analysis was utilized in the qualitative study.

### Results

The overall patient safety culture score was 47.6% (95% CI: 42.7, 52.5). Age (β = 1.196, 95% CI: (0.968, 1.322), patient safety training (β = 0.168, 95% CI: 0.040, 0.297), working in pediatric wards (β = 0.236, 95% CI: 0.099, 0.370), and resource availability (β = 0.346, 95% CI: 0.220, 0.473) were significantly associated with patient safety culture. The in-depth interviews identified infrastructure, communication barriers, lack of management support, poor governance, healthcare professionals' knowledge, skills, and attitudes, and patient involvement during treatment as factors affecting patient safety.

**Data Availability Statement:** The supporting data for this project is publicly accessible and can be found in the following repositories: • https://doi.

org/10.5281/zenodo.13759397 • https://doi.org/10.5281/zenodo.13759323 • https://doi.org/10.5281/zenodo.13759272 • https://doi.org/10.5281/zenodo.13750439 • https://doi.org/10.5281/zenodo.13749393.

**Funding:** The author(s) received no specific funding for this work.

**Competing interests:** The authors declare that they have no conflict of interests.

**Abbreviations:** AAPH, Adis Alem Primary Hospital; AC, Air conditioning; AHRQ, Agency for Healthcare Research and Quality; CI, Confidence Interval; FHCSH, Felege Hiwot Comprehensives Specialized Hospital; FMOH, Federal Ministry of Health; HCP, Healthcare Professionals; HIV, Human immunodeficiency Virus; HSOPSC, Hospital Survey on Patient Safety Culture; ICU, intensive care unit; No, Number; OPD, Out Patient Department; SD, Standard Deviation; SE, Standard Error; SPSS, Statistical Package for Social Science; PSC, Patient Safety Culture; TGSH, Tibebe Ghion Specialized Hospital; WHO, World Health Organization.

## Conclusions

This study concludes that the patient safety culture in the studied hospitals is suboptimal, falling below the acceptable threshold. Enhancing resource availability, providing continuous patient safety training, improving communication systems, and fostering a supportive management environment are essential steps towards building a safer healthcare system.

## Introduction

Patient safety is a global concern at all levels of healthcare systems, with the primary goal of reducing risks to patients during the provision of healthcare services [1]. Today, patient harm resulting from unsafe care is a significant and growing public health challenge worldwide, ranking among the leading causes of death and disability [2]. Evidence indicates that 10% of patients are harmed while receiving healthcare, with at least 50% of these cases being preventable [3]. A culture of patient safety involves organized actions that establish values, beliefs, and behaviors within healthcare settings, aimed at prioritizing patient safety, reducing the occurrence of preventable harm, and minimizing the impact of injuries when they do occur [1].

Globally, patient harm due to unsafe healthcare practices ranks as the 14th leading contributor to the global burden of disease [3]. Medication errors, unsafe surgical procedures, healthcare-associated infections, unsafe injection practices, diagnostic errors, unsafe transfusion practices, radiation errors, and venous thromboembolism are common medical practices that affect patient safety [4–6].

According to studies conducted in Iran, Vietnam, China, India, Afghanistan, and Ghana, the overall levels of patient safety culture were 62.9%, 74.2%, 76.0%, 58.0%, 44.0%, and 58.1%, respectively [7–12]. In Ethiopia, studies conducted on healthcare professionals in Bale, Jima, Addis Ababa, Wollega, Dessie, and Gondar showed that the overall patient safety culture ranged from 36.8% to 49.2% [13–18].

Patient safety grades assess the quality of safety within healthcare settings by evaluating how well facilities protect patients from preventable harm, such as medical errors, infections, or accidents, during their care. Patient safety grades are typically reported in categories such as "excellent," "very good," "good," "fair," or "poor," and they help to identify areas where a healthcare facility excels or needs improvement [19, 20]. Previous studies on patient safety in Ethiopian hospitals have reported levels of overall patient safety grades, with ratings of "excellent" and "very good" with 29.3% in Gondar, 34.0% in Jima, 35.7% in Addis Ababa, and 38.3% in Bale Zone hospitals [15, 18, 21].

Medication errors are a significant patient safety concern in Ethiopian hospitals, where at least one out of every two prescriptions is incorrectly written or administered. The burden of medication administration and prescription errors was 58.4% and 55.8%, respectively [5, 22].

Evidence suggests that the majority of preventable errors in hospitals are the result of systemic issues. Communication barriers, organizational knowledge transfer, insufficient information flow, reporting systems, training, staffing patterns, workflow, and technical failures are all controllable factors for patient harm [12, 18, 23, 24].

Studies have revealed that socio-demographic characteristics such as age, gender, work experience, and educational status significantly affect patient safety [15, 25, 26]. Regarding work-related factors, participation in the patient safety programs, reporting adverse drug reactions, and patient involvement are important factors that influence patient safety [13, 21].

Despite significant investments made to improve patient safety, the implementation gap remains a long-standing concern in Ethiopia's healthcare system [27]. While most low- and

middle-income countries face substantial challenges due to limited resources, these concerns are particularly pronounced in Ethiopia [28, 29].

Despite the recognized importance of mixed methods studies in providing a thorough understanding of safety culture, there remains a significant gap in practical, context-specific evidence concerning patient safety culture and its influencing factors within the Ethiopian healthcare system. One of the ultimate goals of the Ethiopian National Health Care Quality Strategy is to continually ensure and improve patient safety [30]. Therefore, this study aimed to assess patient safety culture and its associated factors among healthcare professionals in Bahir Dar City public hospitals, Northwest Ethiopia.

## Methods and materials

### Study design, area and period

A facility-based cross-sectional study design, triangulated with qualitative methodologies (convergent parallel mixed method), was conducted from March 10 to April 10, 2022, at public hospitals in Bahir Dar City, Northwest Ethiopia. The integrated design allowed the research team to maintain study fidelity with two sequences: a quantitative measurement and a qualitative assessment. The findings from the two segments were unified and triangulated to provide an overall interpretation of the data. The city has one primary and two specialized public hospitals that offer healthcare services across various specialties, including emergency care, gynecology and obstetrics, surgery, pediatrics and child health, internal medicine, ophthalmology, psychiatry, dentistry, pharmacy, laboratory services, radiology, physiotherapy, and other services.

### Inclusion criteria

All healthcare professionals who had worked for at least six months prior to data collection were included.

### Exclusion criteria

Healthcare professionals who were on education, long-term training, or extended leave at the time of the survey were excluded.

### Sample size determination

The sample size for the quantitative study was calculated using a formula for a single population proportion with a 95% confidence level, a margin of error of 0.05 and a proportion of patient safety culture(p = 45.3%) from Gondar Comprehensive Specialized Hospital [18]. After adding a 10% non-response rate, the final sample size was 420.

For the qualitative study, semi-structured, in-depth interviews were conducted based on the principle of saturation, meaning that data collection continued until no new information was obtained from additional interviews. The sample size was determined according to theoretical saturation, the point in data collection when new data no longer provide additional insights into the research questions.

### Sampling technique and procedure

A total of 1,805 healthcare professionals were employed across three public hospitals. A stratified sampling technique was used to select 420 study participants from these hospitals. Healthcare professionals were stratified based on their numbers within each hospital. The number of sample points for each stratum was determined using a proportional allocation formula.

Subsequently, the required sample sizes were selected using a simple random sampling technique within each stratum. As a result, 28 participants were selected from Adis Alem Primary Hospital, 170 from Felege Hiwot Comprehensive Specialized Hospital, and 195 from Tibebe Ghion Specialized Hospital.

A purposive sampling technique was employed to select 9 participants for the qualitative study, focusing on their profession, department, work experience, familiarity with the hospital, and willingness to participate. The study specifically included healthcare professionals with at least five years of work experience to ensure that participants had sufficient expertise and familiarity with the hospital environment.

## Data collection tools and techniques

The Hospital Survey on Patient Safety Culture (HSOPSC) tool, which was adopted from the Agency for Healthcare Research and Quality in the United States (AHRQ) [31], was used to collect quantitative data. The tool was designed to assess hospital staff opinions about patient safety issues, medical errors, and event reporting. It includes 42 items that measure 12 dimensions of patient safety culture: communication openness (3 items), feedback and communication about errors (3 items), frequency of events reported (3 items), handoffs and transitions (4 items), management support for patient safety (3 items), non-punitive response to error (3 items), organizational learning/continuous improvement (3 items), overall perceptions of patient safety (4 items), staffing (4 items), supervisor/manager expectations and actions promoting safety (4 items), and teamwork across and within units (4 items each). All items use the 5-point Likert response scale of agreement (strongly disagree to strongly agree) or frequency (never to always). The tool also includes two questions in which respondents provide an overall grade on patient safety in their work area and the number of events they have reported over the past 12 months.

Data were collected using self-administered questionnaires, with participants completing the questionnaires independently. Three Bachelor of Science-trained nurses were recruited to distribute, guide, and collect the questionnaires; ensuring participants completed them correctly while maintaining confidentiality, while two supervisors oversaw the data collection process.

For the qualitative study, face-to-face, in-depth interviews were conducted using semi-structured, open-ended questions, with additional probing questions employed for more detailed information. The interviews were conducted in Amharic language, with each session lasting between 30 and 50 minutes. The interviews were carried out by two data collectors, including the principal investigator, and focused on factors influencing patient safety culture in hospitals. An ethnographic approach, studying people and their cultures by observing them in their natural environment, was used to explore how patient safety is constructed and maintained within healthcare teams. This research method focuses on the natural environment of healthcare professionals and patients, aiming to provide a detailed understanding of safety practices and challenges through in-depth interviews.

## Data processing and analysis

Data was checked, revised, coded, and entered into Epi-Data version 3.1 before being exported to the Statistical Package for Social Science (SPSS) version 23 for analyses. To ensure completeness, accuracy, and consistency of the data, a discussion session was held daily during the data collection period. The filled-out questionnaires were checked before being received from each data collector.

The overall level of patient safety culture was measured by healthcare professionals' responses to the HSOPSC questionnaire using a Likert scale, with the percentages of positive responses ("strongly agree" and "agree," or "most of the time" and "always") across 12 patient safety culture dimensions (42 items) considered as the overall measure of patient safety culture. We defined scores of 75% and above as indicative of a good patient safety culture or an area of strength. Scores between 50% and 75% were considered to reflect a moderate patient safety culture, while scores below 50% suggest a poor patient safety culture or areas in need of improvement. Negatively worded items were reversed when computing the percent positive response.

Descriptive statistics (frequencies, means, deviations, percentages) were used to summarize socio-demographic, socio-economic, and individual factors. Bi-variable linear regression analysis was performed, and variables with a p-value < 0.20 were included in the multivariable linear regression analyses. Significance was determined at a p-value < 0.05 with a 95% confidence interval (CI). Before fitting the linear regression model, assumptions were verified using scatterplots, histograms, and P-P plots of standardized residuals versus standardized predicted values.

The Durbin-Watson statistics were used to test the assumption of independent errors and autocorrelation, and the result was 1.737, which fell within the acceptable range of 1.50 to 2.50; thus, the assumption of independence was satisfied. The assumption of multi-collinearity was checked using the Variance Inflation Factor (VIF), which showed that the VIF for each independent variable was less than 5. Internal consistency was checked by calculating Cronbach's alpha, which was found to be ($\alpha = 0.896$).

For qualitative data, participants' responses to in-depth interview questions were analyzed using open code software version 4.02, and thematic analysis was conducted on the interview content. The tape-recorded in-depth interviews were transcribed and then translated into English by two professionals fluent in both languages. The translated materials were reviewed multiple times to ensure accuracy and to gain a general understanding of the content. Any discrepancies between the translations were resolved through discussion, and the final version was checked for accuracy and consistency before proceeding with the thematic analysis.

## Ethical considerations

Ethical approval and clearance were obtained from the Ethics and Institutional Review Board (IRB) of the College of Medicine and Health Sciences, Bahir Dar University. A supportive letter was submitted to Adis Alem Primary Hospital, Felege Hiwot Comprehensive Specialized Hospital, and Tibebe Ghion Specialized Hospital to facilitate the study. After explaining the study's purpose and rationale, each participant signed a written informed consent form. All participant data were collected anonymously, and confidentiality was maintained throughout the study.

## Results

### Socio-demographic and socio-economic characteristics of respondents

Of the 420 questionnaires distributed to different departments and units in the three hospitals, 393 completed and valid questionnaires were returned, resulting in a response rate of 93.6%.

Two-thirds of the participants, 260 (66.2%), were male. The respondents' mean age with a standard deviation of 29.66±3.66 years, ranged from 24 to 47 years. Regarding educational status, 267 (67.9%) of the respondents held a bachelor's degree. Among the study participants, 231 (59.5%) of respondents had ≤5 years of work experience (Table 1).

**Table 1. Socio-demographic and socio-economic characteristics of study participants in Bahir Dar City public hospitals, Northwest Ethiopia, 2022 (N = 393).**

| Variables | Category | Frequency | Percent (%) |
|---|---|---|---|
| Sex | Male | 260 | 66.2 |
| | Female | 133 | 33.8 |
| Working institution/Hospital | AAPH | 28 | 7.1 |
| | FHCSH | 170 | 43.3 |
| | TGSH | 195 | 49.6 |
| Religion | Orthodox | 350 | 89.1 |
| | Muslim | 34 | 8.7 |
| | Protestant | 9 | 2.3 |
| Marital status | Single | 205 | 52.2 |
| | Married | 188 | 47.8 |
| Educational status | Bachelor degree | 267 | 67.9 |
| | Specialist doctor | 48 | 12.2 |
| | Resident doctor | 42 | 10.7 |
| | General practitioner | 12 | 3.1 |
| | Master's degree | 12 | 3.1 |
| | Diploma | 12 | 3.1 |
| Profession | Nurse | 178 | 45.3 |
| | Medical doctor | 102 | 26.0 |
| | Midwife | 35 | 8.9 |
| | Pharmacist | 32 | 8.1 |
| | Laboratory technologist | 30 | 7.6 |
| | Anesthetist | 9 | 2.3 |
| | Radiographer | 6 | 1.5 |
| Age (years) | $\leq$29 | 226 | 57.5 |
| | 30–34 | 133 | 33.8 |
| | $\geq$35 | 34 | 8.7 |
| Work experience (years) | $\leq$5 | 231 | 59.5 |
| | 6–10 | 140 | 35.0 |
| | $\geq$11 | 19 | 4.8 |
| Monthly salary in USD($) | $\leq$61.93 | 137 | 34.9 |
| | 61.94–80.17 | 125 | 31.8 |
| | $\geq$80.18 | 131 | 33.3 |

### Work-related and organizational characteristics of study participants

Among the study participants, 69 (17.6%) worked in the surgical department, followed by 54 (13.7%) in the maternity department. The vast majority (94.7%) of participants had direct contact with patients as part of their job. Only 106 (27%) of participants reported having received patient safety training as part of their job. Regarding adverse event reporting, 122 (31.0%) of participants reported experiencing an adverse event in their work. Additionally, only 85 (21.6%) of respondents indicated that they had the appropriate equipment and resources when providing healthcare services (Table 2).

### Patient safety culture

The overall level of patient safety culture was 47.6% (95% CI: 42.7%, 52.5%), which is below the acceptable positive patient safety culture score of 50%.

**Table 2. Work-related and organizational characteristics of study participants in Bahir Dar City public hospitals, Northwest Ethiopia, 2022 (N = 393).**

| Variables | Category | Frequency | Percent (%) |
|---|---|---|---|
| Hospital levels | Primary hospital | 28 | 7.12 |
| | Specialized hospital | 365 | 92.9 |
| Hospital's shifting type | Regular | 258 | 65.6 |
| | Two-shift | 135 | 34.4 |
| Primary work area/ unit/department | Internal medicine | 49 | 12.5 |
| | Surgery | 69 | 17.6 |
| | Pediatrics | 33 | 8.4 |
| | Maternity | 54 | 13.7 |
| | ICU | 42 | 10.7 |
| | Psychiatry | 8 | 2.0 |
| | Emergency | 31 | 7.9 |
| | OPD | 32 | 8.1 |
| | Laboratory | 32 | 8.1 |
| | Radiology | 11 | 2.8 |
| | Pharmacy | 32 | 8.1 |
| Working hours per week | <40 | 34 | 8.7 |
| | ≥40 | 359 | 91.3 |
| Participation in patient safety program | Yes | 114 | 29.0 |
| | No | 279 | 71.0 |
| Adverse event report | Yes | 122 | 31.0 |
| | No | 271 | 69.0 |
| Having direct contact with patient | Yes | 372 | 94.7 |
| | No | 21 | 5.3 |
| Taking patient safety training | Yes | 106 | 27.0 |
| | No | 287 | 73.0 |
| Availability of equipment and materials | Yes | 85 | 21.6 |
| | No | 308 | 78.4 |

Among the 12 dimensions of patient safety culture, 'teamwork within hospital units' had the highest average percentage of positive responses (77.4%), while 'frequency of events reported' had the lowest (29.76%) (Table 3).

## Patient safety grade

Among the study participants, 13.7% and 23.7% rated patient safety as excellent and very good, respectively. Among participants, 37.1% of respondents rated patient safety grade as acceptable, while 20.1% and 5.3% of respondents rated patient safety as poor and failing, respectively (Fig 1).

## Factors associated with patient safety culture

After computing variables and checking assumptions, a bivariate analysis using the linear regression model was conducted for each independent variable against the dependent variable. Variables with a p-value of less than 0.20 were selected for multivariable analyses.

Bivariate linear regression analyses showed that participants' age, marital status, educational status, monthly salary, profession, work experience, patient safety training, participation in patient safety programs, reporting of adverse events, work area, and availability of resources were significantly associated with a positive patient safety culture.

**Table 3. Positive patient safety culture composite levels among healthcare professionals in public hospitals of Bahir Dar City, Northwest Ethiopia, 2022 (N = 393).**

| Patient safety culture dimension | No of Items | Likert scale response rate (%) | | | | | Positive patient safety culture score (%) |
|---|---|---|---|---|---|---|---|
| | | Strongly disagree/ never | Disagree/ rarely | Neutral/ sometimes | Agree/most of the time | Strongly agree/ Always | |
| Teamwork with in units | 4 | 4.1 | 8.1 | 10.6 | 53.1 | 24.3 | 77.4 |
| Manager's expectation &action promoting patient safety | 4 | 4.8 | 26.0 | 26.8 | 35.0 | 7.0 | 42.0 |
| Organizational learning/continuous improvement | 3 | 3.0 | 13.6 | 24.7 | 44.4 | 14.1 | 58.4 |
| Management support for patient safety | 3 | 11.3 | 31.0 | 20.4 | 28.1 | 9.3 | 37.4 |
| Overall perceptions of patient safety | 4 | 5.9 | 24.8 | 21.3 | 38.8 | 9.25 | 48 |
| Feedback & communication about error | 3 | 10.3 | 18.2 | 27.6 | 39.3 | 4.6 | 43.9 |
| Communication openness | 3 | 9.3 | 21.8 | 23.5 | 36.3 | 9.1 | 45.4 |
| Frequency of events reported | 3 | 8.3 | 25.5 | 36.5 | 22.2 | 7.3 | 29.8 |
| Teamwork across units | 4 | 4.9 | 22.6 | 21.1 | 43.4 | 8.1 | 51.5 |
| Staffing | 4 | 7.7 | 21.0 | 23.6 | 37.5 | 10.6 | 48.1 |
| Non-punitive response to errors | 3 | 3.9 | 23.6 | 30.5 | 31.5 | 10.4 | 41.9 |
| Hand off and transition | 4 | 5.6 | 25.8 | 21.7 | 32.2 | 14.1 | 46.3 |
| Total | 42 | 79.1 | 261.8 | 288.1 | 441.8 | 128.0 | 47.6% |

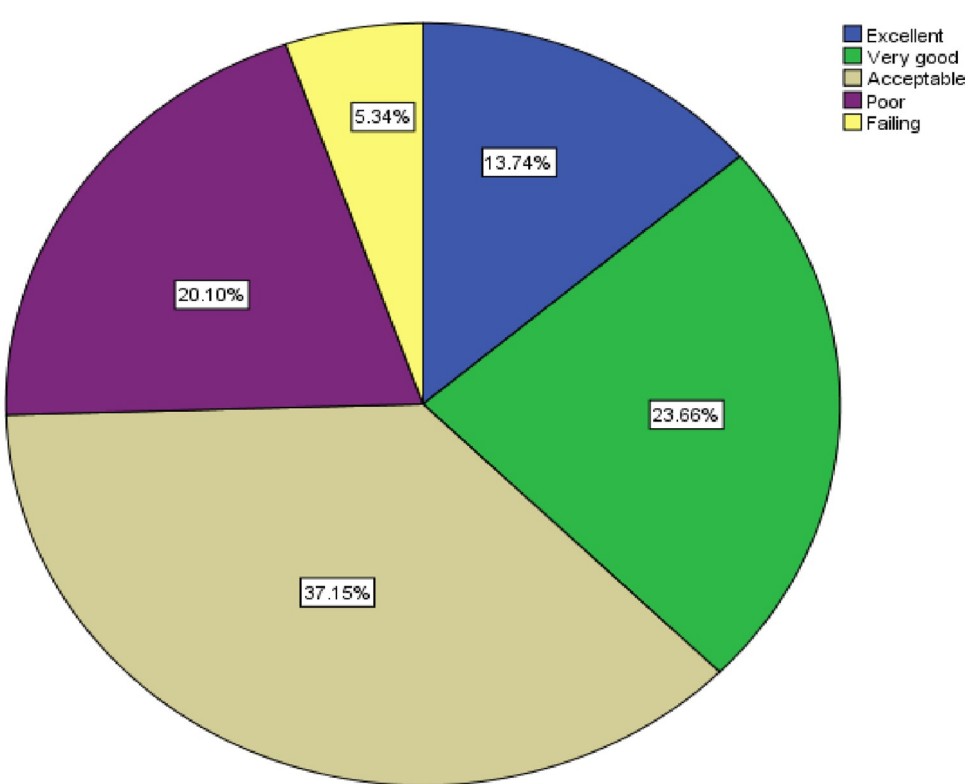

**Fig 1. Hospital's overall patient safety grade among healthcare professionals in Bahir Dar City public hospitals, Northwest Ethiopia, 2022 (N = 393).**

**Table 4. Factors associated with patient safety culture: Multivariable linear regression analysis, Bahir Dar City public hospitals, Northwest Ethiopia, 2022 (N = 393).**

| Variables | Unstand. Coeff. (β) | S.E | p-value | 95% CI of β |
|---|---|---|---|---|
| Age (years) | 1.196 | 0.064 | 0.0001 | (0.968,1.372)* |
| Educational status | | | | |
| Master's degree | 0.235 | 0.139 | 0.119 | (0.167,.345) |
| Specialist doctor | 0.344 | 0.071 | 0.798 | (0.0205,0.484) |
| Marital status | 0.258 | 0.046 | 0.99 | (0.167, 0.347) |
| Monthly salary(USD) | 0.897 | 0.180 | 0.338 | (0.633,1.341) |
| Profession | | | | |
| Medical doctor | 0.120 | 0.054 | 0.619 | (0.014,0.227) |
| Work experience(years) | 0.063 | 0.008 | 0.895 | (0.045,0.076) |
| Work area/department | | | | |
| Pediatric ward | 0.236 | 0.069 | 0.001 | (0.101, 0.371)* |
| Surgical ward | -0.121 | -0.063 | 0.057 | (-0.244,0.003) |
| Participation in patient safety program | 0.162 | 0.062 | 0.562 | (0.112,0 .317) |
| Adverse event report | 0.528 | 0.044 | 0.224 | (0.428, 0. 600) |
| Taking patient safety training | 0.168 | 0.065 | 0.001 | (0.440,0.615)* |
| Availability of equipment and materials | 0 .346 | 0.064 | 0.001 | (0.220,0.473)* |

Notes

*P-value <0.05

Using a stepwise method in the multivariable linear regression analysis, it was found that availability of resources, participant age, working in a pediatric unit, and receiving patient safety training were significantly associated with a positive patient safety culture.

By holding other variables constant, the patient safety culture score increases by 1.196 (β = 1.961, 95%CI: 0.968, 1.322) for each one-year increase in the participant's age. Similarly, healthcare professionals who received necessary equipment and materials on time had a 0.346 increase in their patient safety culture score (β = 0.346, 95% CI: 0.220, 0.473) compared to those who did not receive the necessary equipment and materials.

Controlling for other variables, study participants who received patient safety training showed an improvement in their patient safety culture score by 0.168 (β = 0.168, 95% CI: 0.041, 0.297) compared to those who did not receive patient safety training.

Healthcare professionals working in pediatric wards had an increased patient safety culture score of 0.236 (β = 0.236, 95%CI: 0.099, 0.370) compared with those working in outpatient departments, holding other variables constant (Table 4).

## Qualitative study findings

Nine healthcare professionals participated in the interviews. Of these, four (44.4%) were nurses, and two (22.2%) were doctors. The majority of participants (77.8%) were male, and all had direct patient contact. The average age of the participants was 34 years, ranging from 30 to 42 years. Notably, no new information emerged after the ninth interview (Table 5).

The study identified three major themes influencing patient safety culture, based on healthcare professionals' perceptions: organizational-related factors, healthcare professional-related factors, and patient-related factors. These themes emerged from in-depth interviews conducted with nine healthcare professionals, including nurses, doctors, midwives, and a laboratory technologist, all working in public hospitals in Bahir Dar City, Northwest Ethiopia.

**Table 5. Socio-demographic characteristics of qualitative study participants in Bahir Dar City public hospitals, Northwest Ethiopia, 2022 (N = 9).**

| Variables | Category | Frequency | Percent (%) |
|---|---|---|---|
| Sex | Male | 7 | 77.8 |
| | Female | 2 | 22.2 |
| Religion | Orthodox | 8 | 89.0 |
| | Muslim | 1 | 11.0 |
| Marital status | Single | 3 | 33.3 |
| | Married | 6 | 66.7 |
| Profession | Nurse | 4 | 44.4 |
| | Midwife | 2 | 22.2 |
| | Laboratory technologist | 1 | 11.1 |
| | Doctor | 2 | 22.2 |
| Age(years) | 30–34 | 5 | 55.6 |
| | 35–39 | 3 | 33.3 |
| | ≥40 | 1 | 11.1 |
| Work experience(years) | <10 | 5 | 55.5 |
| | ≥10 | 4 | 44.5 |

## Organizational-related factors

Participants highlighted the lack of resources and equipment as a significant factor compromising patient safety. They identified shortages of essential items such as gloves, linens, safety boxes for needles and sharp materials, laboratory reagents, medical equipment, and related supplies as critical issues affecting patient care.

*". . ..The availability of supplies and medical equipment is essential to establishing patient safety. However, in this hospital, there aren't enough essential supplies and equipment to maintain patient safety. For example, there is only one sphygmomanometer, and there isn't enough equipment like thermometers, glucometers, and stethoscopes in this ward to assess patients and ensure their safety"* (nurse, P7).

The need for safe physical infrastructure and a secure environment was also emphasized by the participants. They described safe infrastructure as comprising strong buildings with adequate space, proper ventilation, cleanliness, and natural lighting. Essential equipment such as wheelchairs, stretchers, patient screens, and beds with side rails were also deemed crucial for ensuring patient safety.

*"Infrastructure itself poses safety challenges, such as inadequate latrines, water supply, showers, room ventilation, and the methods used to transport patients between departments. For example, due to limited space in the neonatal intensive care unit, more than three neonates are treated on the same bed with a single AC machine, while maternal admissions occur on the fourth floor, and neonatal admissions are on the third floor"* (midwife, P9).

Participants also expressed concerns about staffing shortages and the high workloads faced by healthcare professionals, which they believed negatively impacted patient safety. They stressed the importance of balancing the number of healthcare professionals with the patient load to provide safe and effective care.

*". . .In most areas of our hospital, the nurse-to-patient ratio is low. This imbalance forces nurses to spend most of their time in a stressful and overcrowded environment, which is neither comfortable nor conducive to maintaining patient safety"* (nurse, P5).

Participants identified a lack of training and ongoing professional development, along with insufficient guidelines and standard practices, as significant threats to patient safety. They highlighted critical gaps in healthcare systems that jeopardize patient safety, pointing specifically to inadequate training and ongoing professional development, as well as the absence of clear guidelines and standard practices.

*". . .. In order to provide quality healthcare services, healthcare professionals must address gaps in their knowledge, skills, and attitudes through training or continued professional development. However, there is a lack of educational opportunities and insufficient training to support quality healthcare delivery"* (nurse, P1).

*"Guidelines and protocols are essential for healthcare organizations to foster shared knowledge and ensure quality care according to established standards. However, in this hospital, there are no standard protocols to guide staff, such as treatment guidelines, rounding procedures, admission and discharge protocols, medication delivery protocols, or patient transfer protocols. Consequently, decisions are often subjective and inconsistent across departments, which negatively impacts patient safety"* (doctor, P4).

Most respondents emphasized that the hospital lacked a system for reporting adverse events or errors, and that many healthcare professionals were reluctant to report incidents. This lack of a reporting mechanism not only undermines efforts to identify and address potential risks but also fosters a culture of fear and silence among healthcare professionals.

*". . .Many healthcare professionals are unwilling to report adverse events or mistakes because they fear being blamed and held responsible, so they remain silent"* (nurse, P7).

Some respondents pointed to a lack of good governance as a major factor influencing patient safety. Respondents highlighted the critical role of governance in ensuring patient safety, emphasizing that deficiencies in this area are often at the root of many systemic issues in healthcare.

*". . .In my opinion, poor governance is one of the key factors that greatly affects patient safety. For example, most hospital complaints regarding medical equipment, pharmaceuticals, laboratory reagents, human resources, and other issues are not budget-related but stem from poor governance"* (laboratory technologist, P6).

Respondents also mentioned that management support and communication styles with healthcare professionals had a significant impact on patient safety. The majority of participants stressed that to build a strong patient safety culture; managers must collaborate with healthcare professionals to ensure staff satisfaction.

*". . .Managers often act as fault-finders and seem only interested in patient safety when an error or event occurs. There is no clear communication between staff and management, and numerous issues require close discussion and managerial action, such as career development, educational opportunities, training, occupational risk allowances, quality of service, and other organizational problems"* (laboratory technologist, P6).

### Healthcare professional-related factors

Participants identified low levels of knowledge, skills, and attitudes among healthcare professionals as critical factors affecting patient safety. This was particularly evident in cases where insufficient training and inadequate clinical judgment led to serious consequences on patient safety. Such experiences underscore the vital need for healthcare professionals to possess the necessary competencies and attitudes to prevent avoidable harm and ensure the delivery of safe and effective healthcare services.

*". . .Healthcare professionals' knowledge and attitudes directly impact patient safety. I remember a case from a year ago involving a laboring mother who delivered twins via cesarean section. Unfortunately, she experienced postpartum hemorrhage, and her condition went unnoticed. Despite subsequent surgical intervention, she died, and the tragedy was kept quiet"* (midwife, P3).

*". . .Healthcare professionals' knowledge and skills are crucial for ensuring patient safety. They must be adequately trained to deliver safe healthcare services"* (nurse, P5).

Poor communication and inadequate feedback among healthcare professionals were also highlighted as significant issues. Participants pointed out the lack of documentation, ineffective patient handovers, and insufficient communication between patients and caregivers as factors undermining patient safety.

*". . .Communication is one of the major problems we face in our hospital. There is no proper patient handover system, and most healthcare professionals neither communicate effectively nor document their actions. Often, admitted patients are not informed about their condition, the expected length of stay, the likely outcomes, or even the nature of their disease. This lack of communication affects treatment, leading some patients to sign against medical advice or leave the hospital prematurely"* (nurse, P7).

*". . .The exchange of information during shift changes is inadequate. Most healthcare professionals in this hospital lack a well-developed patient handover system, whether in oral or written form"* (midwife, P9).

### Patient-related factors

Respondents indicated that patients' perceptions of their health, their ability or inability to inquire about their rights regarding healthcare services, and their interactions with healthcare staff significantly influence patient safety. Additionally, patients' decisions, awareness of their conditions, and communication with healthcare providers can directly impact the outcomes of their care, sometimes with life-threatening consequences.

*"Patient safety is also influenced by the patients themselves. For example, last year, a patient with chemical burns was admitted to our ward. While receiving treatment, he refused further therapy and signed a medical refusal form. A week later, he returned with severe complications, including septic shock and dehydration. Unfortunately, he died shortly after"* (doctor, P4).

*". . .Patients' awareness and perception of their disease can also impact their safety. When I worked in the surgical ward, a woman arrived with gallstones and had been living with HIV*

*for nearly seven years, though her spouse was unaware of her status. Despite our advice to inform her spouse about her condition, she threatened to commit suicide if we disclosed her status"* (nurse, P2).

Some respondents pointed out that a patient's economic situation plays a crucial role in patient safety. They noted that the rising cost of medical care is a barrier for many, particularly those without community-based health insurance.

*". . .A patient's financial ability to afford medical care is a key factor in their safety. Due to financial constraints, some patients cannot afford necessary treatments, leading them to sign out against medical advice or even abscond from the hospital"* (nurse, P1).

## Discussion

This study found that the overall patient safety culture score in public hospitals in Bahir Dar City was 47.6% (95% CI: 42.7%, 52.5%), indicating a generally poor safety culture with significant room for improvement. This finding is consistent with previous studies conducted in other Ethiopian hospitals, such as Gondar Comprehensive Specialized Hospital (45.3%) and East Wollega Zone public hospitals (49.2%) [16, 18], and slightly higher than results from Addis Ababa regional hospitals (44.0%), Dessie town public hospitals (44.8%), Bale Zone public hospitals (44.0%), and Jima Zone public hospitals (36.8%) [10, 13, 14, 17]. The differences in findings may be attributed to variations in study settings, sample sizes, and institutional factors.

However, the patient safety culture score in this study was notably lower compared to studies conducted in countries like Ghana (58.1%), India (58.0%), Iran (62.9%), China (76.0%), and Vietnam (74.2%) [7–9, 11, 12]. This discrepancy may be attributed to the fact that patient safety culture is a relatively new focus in Ethiopia, whereas those countries have addressed patient safety issues earlier [13]. Additionally, the patient safety culture in developing countries like Ethiopia is often hindered by resource constraints, unlike in more developed nations [32].

Among the dimensions of patient safety culture, "teamwork within hospital units" had the highest positive response rate at 77.4%. This suggests that there are strong collaborative efforts and healthy working relationships within hospital units, similar to findings from Gondar (75.0%), Jima (79.4%), Wollega (77.9%), and Addis Ababa (74.8%) [14–16, 18]. A high score for positive teamwork among hospitals units showed the presence of healthy work relationships, teamwork, and efforts to improve patient safety within the units. However, this score was still lower than those reported in Ghana (83.0%), China (86.9%), and Vietnam (91.3%) [8, 9, 11], indicating that, while teamwork is strong, there remains room for improvement compared to international benchmarks [21].

Conversely, the "frequency of events reported" was the dimension with the lowest positive response rate (29.8%), highlighting a significant area for improvement. Only 26.0% of respondents reported adverse events in the last year, suggesting a low rate of event reporting likely due to fear of reprimand and a lack of safety awareness [21]. This observation is supported by qualitative data where participants expressed that there was no established event reporting system, and healthcare professionals were generally reluctant to report errors.

Management support for patient safety also scored low (37.4%), another area identified for improvement. The lack of managerial commitment and engagement in promoting patient safety was a recurring theme in the qualitative interviews, where participants noted that managers often focused on finding faults rather than fostering a supportive safety culture. Other

studies conducted in Gondar (30.6%), Dessie (34.0%), and Jimma (37.8%) [14, 17, 18] similarly indicate that management support for promoting patient safety is inadequate.

Communication during hospital handoffs and transitions also emerged as a weak area, with a score of 46.3%. This aligns with findings from studies in Jima (41.4%), Dessie (42.4%), and Gondar (44.0%) [14, 17, 18], all of which point to significant communication challenges that could jeopardize patient safety. Participants reported that poor communication and inadequate documentation were prevalent issues, further corroborating these findings.

The perception of staffing adequacy was below the acceptable threshold, with a score of 48.1%. A significant majority (91.3%) of respondents reported working more than 40 hours per week, indicating a potential imbalance between staffing levels and patient load. This concern was echoed in the qualitative interviews, where participants described the stressful and overcrowded work environment as detrimental to patient safety. Previous studies on the Hospital Survey on Patient Safety Culture (HSOPSC) have highlighted significant understaffing among healthcare professionals in various hospital settings: Gondar (47.5%), Addis Ababa (37.4%), Bale (32.0%), Wollega (33.9%), China (37.6%), and Vietnam (49.0%) [8, 9, 13–15, 18]. This chronic understaffing leads to employee fatigue, which in turn increases the risk of medical errors and negatively impacts patient safety. These findings underscore the critical need for addressing staffing issues to enhance patient safety outcomes.

The overall patient safety grade in the present study, where 37.4% of respondents rated it as excellent or very good, aligns closely with findings from previous research conducted in various Ethiopian regions. For instance, studies in Jima (34.0%), Addis Ababa (35.7%), and Bale Zone hospitals (38.3%) reported similar results [13, 15, 21]. This consistency across different studies suggests a broadly comparable level of patient safety perception across these regions.

The present study found that with each additional year of age, participants' patient safety culture scores increased by 1.196 (β = 1.961, 95% CI: 0.968, 1.322), while holding other variables constant. This association may be explained by the fact that as individuals age, they typically gain more experience, develop stronger social interactions, and refine their attitudes and perceptions. These factors contribute to a deeper understanding of patient care, improved ability to prevent errors or adverse events, and a stronger commitment to maintaining patient safety [17, 33].

Healthcare professionals who received the necessary equipment and materials on time when providing care had a 0.346-point increase in their patient safety culture scores (β = 0.346, 95% CI: 0.220, 0.473) compared to those who did not receive the required resources This finding is consistent with studies conducted in Gondar, Ghana, and Afghanistan [10, 11, 18]. This finding was further supported by an in-depth interview, where participants emphasized that the availability of supplies and medical equipment is just as crucial as the role of healthcare professionals in establishing patient safety.

Study participants who received patient safety training had a 0.168-point increase in their patient safety culture scores (β = 0.168, 95% CI: 0.041, 0.297) compared to those who did not receive such training, with other variables held constant. This improvement may be attributed to the general truth that training enhances knowledge, skills, and attitudes, providing essential information on how to uphold a culture of patient safety [34]. This finding was reinforced by insights from an in-depth interview, where participants emphasized the importance of addressing knowledge, skill, and attitude gaps through training or continuous professional development to ensure the delivery of quality healthcare services.

Healthcare professionals working in pediatric wards had a 0.236-point higher patient safety culture score (β = 0.236, 95% CI: 0.099, 0.370) compared to those working in outpatient departments, after controlling for other variables. This may be because treating children often requires greater attention and care, fostering a heightened focus on patient safety [35].

Additionally, maternal and child health services often receive more support from governmental and non-governmental organizations, which may contribute to a stronger patient safety culture in these settings.

## Limitation

This study was conducted only in public hospitals; therefore, the result can not apply to other private hospitals due to potential differences in resources, management practices, staffing, access to advanced technology, and safety protocols. Also, this study included only healthcare professionals, even though patient safety culture may also be significantly influenced by paramedical staff.

The other limitation which was acknowledged in this study was self-reported surveys are subject to bias that respondents are more likely to report what is socially acceptable or preferred.

## Conclusion

The findings revealed that the overall patient safety culture score was 47.6%, which is below the acceptable threshold of 50%. This indicates a relatively poor patient safety culture in the studied hospitals, consistent with similar findings from other Ethiopian hospitals.

Key factors significantly associated with a positive patient safety culture included the availability of necessary equipment and resources, the age of healthcare professionals, patient safety training, and working in pediatric wards.

The qualitative findings corroborated the quantitative results, with healthcare professionals citing organizational challenges, such as resource shortages, inadequate infrastructure, and lack of management support, as major impediments to patient safety. Additionally, issues related to staff workload, communication gaps, and insufficient training were also highlighted as areas needing urgent attention.

The study underscores the need for targeted interventions to improve patient safety culture in Bahir Dar city public hospitals. Enhancing resource availability, providing continuous patient safety training, improving communication systems, and fostering a supportive management environment are essential steps towards building a safer healthcare system.

## Acknowledgments

We would like to thank Bahir Dar University, College of Medicine and Health science, School of Public Health, Department of Health Systems Management and Health Economics for giving us this opportunity to learn and conduct this study. We would also like to thank Tibeb Ghion, Felege Hiwot and Addis Alem hospitals managers for their cooperation during data collection, and data collectors, supervisors and study participants for their active participation to complete questionnaire and voluntariness for in-depth interview.

## Author Contributions

**Conceptualization:** Daniel Atinafu, Gebremariam Getaneh, Getachew Setotaw.

**Data curation:** Daniel Atinafu, Gebremariam Getaneh, Getachew Setotaw.

**Formal analysis:** Daniel Atinafu, Gebremariam Getaneh.

**Funding acquisition:** Daniel Atinafu.

**Investigation:** Daniel Atinafu.

**Methodology:** Daniel Atinafu, Getachew Setotaw.

**Project administration:** Daniel Atinafu.

**Validation:** Daniel Atinafu.

**Writing – original draft:** Daniel Atinafu.

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
