## [Decision Letter · Decision Letter 0]

8 Apr 2024

PONE-D-24-03486Assessment of patient safety culture and associated factors among healthcare professionals in Bahir Dar city public Hospitals, Northwest Ethiopia: mixed study designPLOS ONE

Dear Dr. Atinafu,

Thank you for submitting your manuscript to PLOS ONE. After careful consideration, we feel that it has merit but does not fully meet PLOS ONE’s publication criteria as it currently stands. Therefore, we invite you to submit a revised version of the manuscript that addresses the points raised during the review process.

We look forward to receiving your revised manuscript.

Kind regards,

Lovenish Bains, MS, FNB, FACS, FRCS (Glas), FICS, FIAGES

Academic Editor

PLOS ONE

Journal Requirements:

2.  In the online submission form, you indicated that "With the request of correspondence author"

3. Please include a caption for figure 1.

4. Please ensure that you refer to Figure 1 in your text as, if accepted, production will need this reference to link the reader to the figure.

5. Please upload a copy of Figure 3, to which you refer in your text on page 11. If the figure is no longer to be included as part of the submission please remove all reference to it within the text.

6. Please include your tables as part of your main manuscript and remove the individual files. Please note that supplementary tables (should remain/ be uploaded) as separate ""supporting information"" files

**Additional Editor Comments:**

This mixed-methods study investigates patient safety culture in public hospitals of Bahir Dar, Ethiopia.

While the research topic is relevant, the manuscript requires substantial improvements and revisions before any consideration.

The major areas are 1.Writing and structure require attention, 2. Needs work on reporting and analysis, 3. Discussion section could be strengthened.

Further, the grammar, syntax and inconsistencies needs to be checked at multiple statements.

Various other comments and as derived from reviewer feedback are:

Abstract and introduction could benefit from focus and clarity. HSOPS composite scores and survey administration details are missing. Statistical analysis methods raise concerns about outcome dependence. De-identification of interview quotes could be improved.

Sampling method and representativeness criteria need elaboration. Rationale for different experience requirements in quantitative vs qualitative parts is unclear.

Interview guide, coding process, and thematic analysis details are missing.

Results section, particularly qualitative findings, needs streamlining.

Avoid repetition from other sections. Focus on data triangulation and proper referencing. Clarify new hypotheses presented. Tables and figures require improvement. Differentiate patient safety from patient safety culture.

Explain the method triangulation approach mentioned in the abstract.

Define "safety box."

Reviewers' comments:

Reviewer's Responses to Questions

**Comments to the Author**

1. Is the manuscript technically sound, and do the data support the conclusions?

Reviewer #1: Yes

Reviewer #2: Partly

Reviewer #3: No

Reviewer #4: Partly

2. Has the statistical analysis been performed appropriately and rigorously? 

Reviewer #1: Yes

Reviewer #2: No

Reviewer #3: I Don't Know

Reviewer #4: I Don't Know

3. Have the authors made all data underlying the findings in their manuscript fully available?

Reviewer #1: Yes

Reviewer #2: No

Reviewer #3: No

Reviewer #4: Yes

4. Is the manuscript presented in an intelligible fashion and written in standard English?

Reviewer #1: No

Reviewer #2: Yes

Reviewer #3: No

Reviewer #4: No

5. Review Comments to the Author

Reviewer #1: A pleasure to review your manuscript. The manuscript requires general editing for the language to make it easier for the readers to understand. Also please include the limitations of your study and how they were overcome in order for the results to be considered as more generalizable.

Reviewer #2: This is well-written and interesting, mixed methods paper that evaluates patient safety culture in hospitals in several locations in Ethiopia. Patient safety culture was evaluating using both the Hospital Survey on Patient Safety Culture (HSOPS; referred to in the paper as HSOPSC because of the trademark for HSOPS) survey and narrative responses. The combination of both is a huge strength, the free response information provides context and details that are not necessary captured by the structured survey. My major comments are adding information about the HSOPS results, adding details about the survey administration, removing outcome dependent reporting using p-values, and better de-identification of information in respondent comments.

Major Comments

1) Graph/table of composite scores w/means and SDs/SEs/etc: Typically, the composites scores (e.g., Teamwork, Organizational Learning) for this instrument are presented, not just the overall patient safety culture score. I recommend reporting all of the composites because those tend to be more "actionable" then the overall score.

2) Questions about survey administration:

- It sounds like the survey was administered in a language other than English? Was the current, or a previous, SOPS translation guide followed?

https://www.ahrq.gov/sites/default/files/wysiwyg/sops/surveys/Translation-Guidelines-SOPS-090222.pdf

- Did the authors obtain permission to use the survey internationally (ideally, this is done before translation and survey administration but it can be done later): Email: SafetyCultureSurveys@westat.com to request AHRQ’s permission to translate and use the surveys internationally.

- Which version of HSOPS was used? I'm assuming it was 1.0 because reference 34 was published in 2016, before version 2.0 came out.

3) Selection using statistical significance and stepwise regression is problematic because it's outcome dependent reporting, which is circular- this is nicely summarized as inflated effect sizes in Vul et al. 2009. On p. 11, the manuscript states variables "... with p < 0.20 were selected for multivariable analysis." and stepwise regression was used, which also selects variables based on stat significance. I recommend including all results and not using p-values for selecting inclusion, otherwise results are outcome dependent which is circular (see https://amstat.tandfonline.com/doi/full/10.1080/00031305.2016.1154108#.Vt2XIOaE2MN [supplemental material]). In more detail, this issue is described as follows:

"Less often stated is the even more crucial assumption that the analyses themselves were not guided toward finding nonsignificance or significance (analysis bias), and that the analysis results were not reported based on their nonsignificance or significance (reporting bias and publication bias). Selective reporting renders false even the limited ideal meanings of statistical significance, P-values, and confidence intervals." (p. 8 of the supplemental material linked above). Also, the section stating p-values shouldn't be used as a bright line criterion.

Similarly, stepwise regression is flawed because results are outcome dependent (see https://link.springer.com/article/10.1186/s40537-018-0143-6)

It may be helpful to consult with a statistician. These are very common issues, not just in health science.

4) Better de-identification of comments: I strong recommend removing ages and probably staff position too from the comments. Maybe the unit too? My concern is that it may be possible to identify specific individuals with just three pieces of information: age, staff position, and ward. Or at least narrow it down to a handful of individuals.

Minor Comments

1) p-values: There are several p-values that are reported as .000. This is picky, but reporting p-values as zero is generally incorrect (see Wu et al. 2020) other than a few exceptions. My guess is SPSS is rounding a very small number to .000. It would be better to report these very small p-values as p < .001.

2) Data sharing: If possible, I suggest sharing de-identified aggregated data on OSF (https://osf.io/) or a similar platform. Data could be aggregated by hospital. Please be very, very careful to not sharing any individual level respondent data (such as age) that could be used identify specific individuals.

References

Wu, Y., Zhou, C., Wang, R., Ye, X., Yang, L., Li, C., ... & Cong, W. (2020). Statistical reporting in nursing research: addressing a common error in reporting of p values (p=. 000). Journal of Nursing Scholarship, 52(6), 688-695.

Vul, E., Harris, C., Winkielman, P., & Pashler, H. (2009). Puzzlingly high correlations in fMRI studies of emotion, personality, and social cognition. Perspectives on Psychological Science, 4(3), 274-290.

Reviewer #3: Dear authors,

thank you very much for the opportunity to read your manuscript. Patient safety and patient safety culture are a topic of utmost importance; thus, I do think the manuscript is relevant. Still there is several aspects that I suggest to be rewritten.

General aspects:

Proofreading would be very helpful as there is grammar and spelling mistakes and also inconsistent writing styles e.g. healthcare Professionals vs healthcare professionals, healthcare vs health care vs health-care, work flow, patient safety vs. patient-safety etc., “Medication error is A patient safety concern…”. Generally, the language level is currently quite poor. Sometimes there are too many spaces in individual sentences, which decreases readability. Please have your work corrected by a professional proofreader.

The overall work could profit from a new and revised structure.

Abstract:

The sentence describing the objective is almost exactly the same as the second sentence in the background. I would use the space to insert relevant information in the background section instead of reposting a sentence. The paragraph of the results includes too many details which makes it hard to read. I would keep it more focused in the abstract. Also, are the identified factors affecting patient safety or the patient safety culture. I would be very careful with the use of the words and not mix them up.

Introduction:

The first sentence quotes the WHO. Are you sure you are referring to an information provided by WHO by saying that patient safety is essential due to an increase in patient injury? The connection between increasing patient injuries and the essential need for patient safety would be new to me.

Before listing the patient safety scores of different countries, please elaborate in more detail how the score is measured and how it can be interpreted. I would reduce the number of mentioned examples in the introduction e.g. the sources 24 and 25 do not present an important insight as there are only stated and not further explained or connected to the previous data. This paragraph also has linguistic shortcomings that impair readability e.g “Clinical handover is ongoing problems that affect patient safety culture in healthcare delivery system”. The last sentence is unclear as it states that “this study attempted to analyze the extent of patient safety culture …and associated factors…in order to achieve this goal”. Please rephrase this sentence, as it currently makes no sense.

Methods:

The authors mention that they used a simple random sampling technique to select a representative sample. Please elaborate in more detail on what the representativeness was based e.g. gender, discipline, background age? By explaining your strategy of representativeness, I would have understood that you used quotas and not a lottery to select participants.

Why didn’t you address all healthcare professionals in the respective hospitals? Most of the participants of the quantitative questionnaire have work experience of less than 5 years. Why did you select interview participants for the qualitative interview that have more than 5 years of experience? I argue that a triangulation of data is more difficult here, as you focus on different target groups. How did the participants get access to the questionnaire? Was it online or paper-based? Please elaborate in more detail.

Please provide the interview guide that was used for interviewing the 9 healthcare professionals and elaborate more on the methodology used. Who coded the transcripts? Was the method of “Thematic Analysis” conducted? How many codes where created? What are the names of the codes.

In your paragraph about the ethical considerations, you mention a supportive letter. Why did you need this when you had an ethical approval?

Results

In general, the results section does not yet have a common thread. Currently the qualitative parts ready more like a list of quotes and not a structured report on findings based qualitative methods. Please streamline the results section and adhere more strictly to qualitative research methods.

Results/ Patient Safety Grade: I would also mention the biggest part that answered with acceptable to give the whole picture.

Discussion:

What is meant my the second sentence “The current study discovered that hospitals have a low patient safety culture based on the overall positive response rate for all dimensions of the HSOPSC survey.”? This reads like a very general statements but I guess you only want to talk about the specific hospitals you looked into.

The discussion section starts with a repetition of data that was already mentioned in the introduction section. I suggest shortening it and only focus on data not referred to before. Please focus more on really triangulating your quantitative and qualitative data.

Furthermore, besides the patient safety scores in different countries that are cited correctly, other text paragraph parts are missing a proper reference. E.g. “Another possible reasons is study settings…..” The whole paragraph of assumptions does not link to references or own findings. Also, here major language flaws appear e.g. “a high score for positive teamwork….is teamwork” and make it difficult to comprehend the intended meaning. In the discussion a lot of new hypotheses are stated e.g. “This revealed that event reporting culture of HCP was low. The reason behind this could be…”. Reading the introduction and aim of the study I assumed the qualitative interviews would ask exactly for those aspects.” It is therefore important to clarify once again what exactly this study has shown - especially with regard to the study objective stated in the introduction.

Tables & figures

Table 2: Work area: The different unit descriptions remain unclear. What is a department called “Medicine”?

Table 1: The use of upper and lower case letters is inconsistent

Figure 1: mix up with figure 3 (results section)

Minor comments:

What is a “safety box”? (Mentioned on page 12), What is meant by “essential medicine”? MedicineS? Or something else – please clarifty. What is meant by “low attitude” – please explain in more detail!

Reviewer #4: The topic of the article is interesting. However, a clearer distinction needs to be made between patient safety and patient safety culture. They are not the same concepts. However, they are repeatedly equated in the article. This aspect should be emphasized more strongly.

A method triangulation is mentioned in the abstract. However, the exact nature of this triangulation is not explained in the paper.

Introduction

The logical sequence in the first sentence is not clear to me. What is meant by "individual and group value"? To what extent are treatment errors "common medical practices"? What is a patient safety culture? More explanation is needed here - also to understand what an average patient safety culture is and how the figures quoted should be categorized. I would rank these figures from lowest to highest or only give min. and max. values. What is the "general perception of patient safety culture"? Who perceives it? What does the "positive response rate" mean? What does the prevalence in the 3rd paragraph (57.6%) refer to?

How does the 4th paragraph relate to the topic of the article?

What do you mean by "local capacity"?

What is the aim of the study?

Methods and Materials

Which qualitative method was used? More detailed information and the relevant literature sources are missing here.

Where did the respondents in the quantitative part of the study work in order to fulfil the inclusion criteria?

Is the calculation (formula) of the sample size needed in the paper?

How large was the population of the quantitative survey?

What is the sample representative of? How does a lottery sample fit in with representativeness? What did potential study participants register for?

Why were people with at least five years of experience interviewed for the qualitative survey, but six months is sufficient for the quantitative survey?

Why was the survey adapted?

Why is the number of items listed for some topics but not for others? How was the survey conducted and the data collected (paper-pencil, online, self-completion, survey by whom?)?

Is Amharic the local language? Were the qualitative interviews conducted by only one person? How many researchers worked with the data in total? Why were the transcripts translated into English? The method should be described in more detail, how was it analyzed, what was "evaluated and categorized" and how? Who discussed the data and which data exactly (qualitative or quantitative)? What was checked before the questionnaires were collected? Why was the questionnaire tested in Gondar Hospital?

Ethical Considerations

What is meant by the "supportive letter", what is its reason? Were informed consent forms completed for both the qualitative and quantitative surveys?

Results

The sample appears to be very homogeneous in terms of age. The description of the socio-demographic data is somewhat confusing, the selection of information given in the text seems arbitrary. Is the sample representative of the population in terms of gender, age, religion, work experience and marital status? What role does marital status play in the topic and why is it mentioned?

The logical structure of the characteristics of the study participants is not coherent.

The chapters "Magnitude of Patient Safety Culture" and "Patient Safety Grade" should be explained more (possibly in the introduction) - more background is needed here: what does this mean? what is acceptable,...?

In my opinion, the description of the regression model (dependent/independent variable) is not necessary. More interesting would be: What was the dependent variable?

What differences are there with regard to the influencing factors?

The description of the qualitative results lacks a common thread - it is just a string of quotes. The storyline should be worked out more clearly.

The handover system, which is described in the qualitative results as "healthcare professional related factors", is actually an organizational problem. The "patient related factors" are a strange category - many of the points described are actually tasks of health care professionals: communication, education, empowerment,... or organizational tasks.

Discussion

Very long and repetitive of what is already in other parts of the paper (sometimes even the same quotes). The numbers mentioned are disjointed and need to be embedded. What do you mean by "sophisticated reason" (3rd paragraph)? Where are the references for the hypotheses put forward?

I lack clarity regarding the aim of the study and also the results.

The list of abbreviations is not complete.

The tables should be more informative, e.g. hospital levels (Tab 2), what is meant by: primary work area/dept./unit: medicine?

The article needs a proof-reading, some passages are difficult or impossible to understand. The indication of the standard deviation is unusual for me - this should be indicated as +/-. I would have liked to have seen the questionnaire and the guidelines for the quantitative and qualitative survey. This would perhaps lead to more clarity.

6. PLOS authors have the option to publish the peer review history of their article (what does this mean?). If published, this will include your full peer review and any attached files.

Reviewer #1: No

Reviewer #2: No

Reviewer #3: No

Reviewer #4: No

---

## [Author Response · Author response to Decision Letter 0]

18 Jun 2024

We have addressed the comments accordingly.

---

## [Decision Letter · Decision Letter 1]

12 Aug 2024

PONE-D-24-03486R1Assessment of patient safety culture and associated factors among healthcare professionals in Bahir Dar city public Hospitals, Northwest Ethiopia: mixed study designPLOS ONE

Dear Dr. Atinafu,

Thank you for submitting your manuscript to PLOS ONE. After careful consideration, we feel that it has merit but does not fully meet PLOS ONE’s publication criteria as it currently stands. Therefore, we invite you to submit a revised version of the manuscript that addresses the points raised during the review process.

Please include the following items when submitting your revised manuscript:A rebuttal letter that responds to each point raised by the academic editor and reviewer(s). You should upload this letter as a separate file labeled 'Response to Reviewers'.A marked-up copy of your manuscript that highlights changes made to the original version. You should upload this as a separate file labeled 'Revised Manuscript with Track Changes'.An unmarked version of your revised paper without tracked changes. You should upload this as a separate file labeled 'Manuscript'.

We look forward to receiving your revised manuscript.

Kind regards,

Lovenish Bains, MS, FNB, FACS, FRCS (Glas), FICS, FIAGES

Academic Editor

PLOS ONE

Reviewers' comments:

Reviewer's Responses to Questions

**Comments to the Author**

1. If the authors have adequately addressed your comments raised in a previous round of review and you feel that this manuscript is now acceptable for publication, you may indicate that here to bypass the “Comments to the Author” section, enter your conflict of interest statement in the “Confidential to Editor” section, and submit your "Accept" recommendation.

Reviewer #3: (No Response)

Reviewer #4: (No Response)

2. Is the manuscript technically sound, and do the data support the conclusions?

Reviewer #3: Partly

Reviewer #4: Yes

3. Has the statistical analysis been performed appropriately and rigorously? 

Reviewer #3: Yes

Reviewer #4: Yes

4. Have the authors made all data underlying the findings in their manuscript fully available?

Reviewer #3: No

Reviewer #4: No

5. Is the manuscript presented in an intelligible fashion and written in standard English?

Reviewer #3: No

Reviewer #4: No

6. Review Comments to the Author

Reviewer #3: Dear authors,

thank you very much for the opportunity to read the revision of the manuscript. I think the manuscript clearly improved in the revision. Still, several remarks need to be considered.

General remarks:

• Language improved significantly, please still, check again for typos etc. as they still very frequent in all parts of the manuscript which decreases readability. Often punctuation marks etc. are missing and particularly in some newly written parts the language is not accurate. I get the feeling that the proofreading was done on the old version as some parts are improved now but it seems as if several changes have been made after this proofreading. Those texts clearly need improvement!

• Please check for continuous format style! Sometimes subheadings are formatted in capital letters and sometimes not.

Abstract

• Improved, still proofreading needed.

Introduction:

• Patient safety culture –> capital letters or not?

• “Globally, patient harm attributed to unsafe healthcare delivery is one of the major contributor for the global burden of disease ranking fourteen”  I don’t understand the meaning of this sentence.

• The following sentence is used twice in exactly the same way in the abstract AND in the introduction. Please rewrite: Despite the need for mixed methods studies to achieve a deeper understanding of safety culture, there are few studies providing practical evidence of patient safety culture and associated factors in Ethiopia.

Method

• In general the methods section improved

• I don’t understand the description of the “Inclusion criteria” – the part “for quantitative study” seems to not fit in here?

• What is meant by data collection thought self-administered data collection techniques? Please elaborate.

• Please rewrite the following paragraphs as it lacks clarity and contains several language mistakes: “The interview were conducted by two data collectors including principal investigator and focused on factors influencing patient safety culture in hospitals. For qualitative study, Ethnographic approach was used to explore how patient safety is constructed and accomplished within healthcare teams. This qualitative research method focuses on the natural environment of healthcare professionals and patients, aiming to provide a detailed understanding of safety practices and challenges through in-depth interview.”

• The description of the section “measurement” is confusing. Who defines that a score of 75%+ is considered good? The authors or other studies? Also, the final paragraph is not easy to understand.

• “A tape recorded in-depth interview was transcribed and then translated into English by two professionals” � I don’t understand what is meant? Was it one single tape of one interview or was that the standard procedure for alle interviews? Please elaborate and rewrite.

• The subheadings do not match the text. Why is one subheading called “Data Management and Analysis” and the following “Data Quality Management”? in some parts it remains unclear whether the quantitative or the qualitative data is referred to.

Results

• Why do you have more than two thirds of male respondents? Is that representative for the hospital staff?

• The monthly salary is given in Ethiopia Birr – to increase readability on an international level it would be interesting to have numbers in dollars or rather a categorization whether this income can be considered as average/high/low?

• What is the patient safety grade?

• Qualitative findings: 4 nurses, 2 physicians, what about the rest?

• In the same paragraph there are two versions how to write healthcare professional: “participants expressed that shortage of staff and high workloads on health care professionals negatively affect the patient safety. They believed that number of healthcare professionals in hospital and people served should be balanced to provide safe healthcare service“. Please align!

• The section on “organizational related factors” reads quite unstructured and as a collection of quotes rather than a solid listing of findings. The same is true of the presentation of the “healthcare professional related factors”. Maybe an illustration or table presenting the findings in an overview would help to introduce the sections before stating the quotes.

Discussion

Please focus more on really triangulating your quantitative and qualitative data. As already argued in my comments of the previous review the discussion is lengthy and not on point. What are the main topics you discovered and how are they related to the previous research?

Limitation

Why might the study not be compared to private hospitals? What might there be different and why?

Again, another variation of writing healthcare, this time health-care -please check this throughout the whole paper is demonstrates lack of perfectionism and clarity.

Reviewer #4: Dear authors,

Thank you very much for your revisions and reactions to the comments. Many things have become clearer as a result.

Regarding my comments on the above questions: the data on which your study is based is still missing - but I know that you have indicated that it will be made available in a publication. I have therefore marked ‘no’ as the answer for the time being. The second ‘no’ refers to standard English. I can see that the text has improved a lot compared to the previous version, but there are still some sentences and parts of sentences that do not correspond to standard English. There is also still work to be done on the uniformity of spacing, formatting of upper and lower case letters (e.g. for the interviewees p7/P7/p 7/ P 7 or nurse/Nurse,...).

About the individual chapters:

Abstract: the contents of the conclusion are already in the results - the information is duplicated here

Introduction:

2nd paragraph: meaning unclear - ‘...the global burden of disease ranking fourteen’

3rd paragraph: What is the logic behind the enumeration of the level of Patient Safety Culture? The numbers appear disorganised.

Methods and Materials:

Sample Size Determination:

2nd paragraph: What is meant by ‘saturation theory’? I am only familiar with theoretical saturation, which is mentioned in the next sentence anyway.

Data Collection Tools and Techniques:

2nd paragraph: here it would also be good to include more information in the text about the questionnaires (paper-pencil) and the role of the nurses who ‘collected’ the data.

3rd paragraph: the ‘ethnographic approach’ still seems too broad to me as a qualitative method (cf. ‘statistical methods were used...’ without specification: which ones) - in any case, a source is needed here.

Data Management and Analyses:

5th paragraph: were all interviews recorded and transcribed or just one? I guess it were all, but this is not clear from the sentence.

Results:

Table 1: what is the logic in the enumeration of ‘Educational Status’ and ‘Profession’?

Factors Associated with Patient Safety Culture:

Here is another Table 1 - consistent numbering necessary!

Also: why is marital status relevant?

Discussion:

1st paragraph: What is meant by ‘...and areas with the most potential form improvement’? In this paragraph, too, the logic of the ranking of the results of other studies is not clear to me.

4th-6th paragraph: the quotes from P6, P7 and P5 are also (pretty much) word-for-word identical in the Results (actually they should be identical if they are the same... still I argue that these quotes shouldn't be doubled in the text - I'm sure you have other/more quotes strengthening your argument than just these).

9th and 10th paragraphs: Can you cite sources that support your hypotheses (‘This might be due to the general truth that training increases one's knowledge,....’ and ‘The possible reasons could be the fact that treating children needs more attention and care than treating adults.’).

Conclusion:

Still very weak and repetitive. What are the key findings of your study?

7. PLOS authors have the option to publish the peer review history of their article (what does this mean?). If published, this will include your full peer review and any attached files.

Reviewer #3: No

Reviewer #4: No

---

## [Author Response · Author response to Decision Letter 1]

13 Sep 2024

we addressed all comments accordingly.

---

## [Editor Report · Decision Letter 2]

22 Sep 2024

PONE-D-24-03486R2Assessment of Patient Safety Culture and Associated Factors among Healthcare Professionals in Public Hospitals of Bahir Dar City, Northwest Ethiopia: A Mixed-Methods StudyPLOS ONE

Dear Dr. Atinafu,

Thank you for submitting your manuscript to PLOS ONE. After careful consideration, we feel that it has merit but does not fully meet PLOS ONE’s publication criteria as it currently stands. Therefore, we invite you to submit a revised version of the manuscript that addresses the points raised during the review process.

We look forward to receiving your revised manuscript.

Kind regards,

Lovenish Bains, MS, FNB, FACS, FRCS (Glas), FICS, FIAGES

Academic Editor

PLOS ONE

Journal Requirements:

Additional Editor Comments:

**Dear Authors**,

Do highlight the changes or revisions done in the revised manuscript and ensure a point-by-point response to the reviewers for further assessment.

Do not put track changes or strikethrough manuscript.

**Best**

---

## [Author Response · Author response to Decision Letter 2]

3 Oct 2024

we addressed all comments accordingly

---

## [Decision Letter · Decision Letter 3]

23 Oct 2024

Assessment of Patient Safety Culture and Associated Factors among Healthcare Professionals in Public Hospitals of Bahir Dar City, Northwest Ethiopia: A Mixed-Methods Study

PONE-D-24-03486R3

Dear Dr. Atinafu,

We’re pleased to inform you that your manuscript has been judged scientifically suitable for publication and will be formally accepted for publication once it meets all outstanding technical requirements.

Kind regards,

Lovenish Bains, MS, FNB, FACS, FRCS (Glas), FICS, FIAGES

Academic Editor

PLOS ONE

Additional Editor Comments (optional):

Reviewers' comments:

Reviewer's Responses to Questions

**Comments to the Author**

1. If the authors have adequately addressed your comments raised in a previous round of review and you feel that this manuscript is now acceptable for publication, you may indicate that here to bypass the “Comments to the Author” section, enter your conflict of interest statement in the “Confidential to Editor” section, and submit your "Accept" recommendation.

Reviewer #3: All comments have been addressed

2. Is the manuscript technically sound, and do the data support the conclusions?

Reviewer #3: Yes

3. Has the statistical analysis been performed appropriately and rigorously? 

Reviewer #3: Yes

4. Have the authors made all data underlying the findings in their manuscript fully available?

Reviewer #3: Yes

5. Is the manuscript presented in an intelligible fashion and written in standard English?

Reviewer #3: Yes

6. Review Comments to the Author

Reviewer #3: Dear authors,

thank you very much for the opportunity to read the revision manuscript. I think the manuscript clearly improved in the revision. It still needs a final proof reading but otherwise it is ready for publication. Eg. Introduction: In the second paragraph the text starts with a “.” before “Globally” that needs to be deleted. Also there is no space between “year” and the source (1) in the same sentence. This happens more often – please check for coherence! The second sentence starts identical with the first sentence (7 words identical!) which should changed.

7. PLOS authors have the option to publish the peer review history of their article (what does this mean?). If published, this will include your full peer review and any attached files.

Reviewer #3: No

---

## [Editor Report · Acceptance letter]

28 Oct 2024

PONE-D-24-03486R3 

PLOS ONE

Dear Dr. Atinafu, 

I'm pleased to inform you that your manuscript has been deemed suitable for publication in PLOS ONE. Congratulations! Your manuscript is now being handed over to our production team.

Kind regards, 

on behalf of

Dr. Lovenish Bains 

Academic Editor

PLOS ONE